# Inelastic X-ray Scattering as a Probe of Terahertz Phonon Propagation in Nanoparticle Suspensions

Alessandro Cunsolo

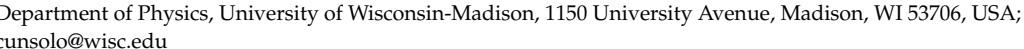

Department of Physics, University of Wisconsin-Madison, 1150 University Avenue, Madison, WI 53706, USA; cunsolo@wisc.edu

**Abstract:** This paper reviews recent inelastic X-ray scattering investigations of simple inhomogeneous materials, such as nanoparticle suspensions in liquids. All studies reported emphasize the ability of immersed nanoparticles to dampen or attenuate acoustic waves through the hosting medium, the effect becoming even more pronounced upon freezing. Additionally, the results show that suspended nanoparticles can cause the onset of non-acoustic modes in the system. Also, the crucial role of Bayesian analysis in guiding spectral line shape modeling and interpretation is discussed. In summary, the presented results demonstrate that the simple inclusion of a sparse amount of nanoparticles profoundly influences sound propagation through a medium. This finding can inspire new avenues in the emerging field of terahertz acoustic steering and manipulation.

**Keywords:** inelastic X-ray scattering; phonon propagation; phononics

## 1. Introduction

In recent years, the interest in the manipulation of acoustic propagation through the control of the mesoscale structure has been growing steadily. This task is especially compelling at terahertz frequencies, where phonons are the leading heat carriers in insulators, and their control becomes crucial to implementing heat flow management [1]. In principle, phononic crystals (PCs) are excellent candidates to achieve this goal [2], as their tailored structure impacts sound propagation, potentially enabling its complete shaping. It was demonstrated that PCs can pave the way for the development of novel thermal devices, such as thermal diodes [3], thermoelectrics [4], and thermocrystals [5]; furthermore, they lend themselves to applications as relevant as energy harvesting [6] and the management/control of thermal properties [7]. A prerogative of these devices is the ability to forbid acoustic propagation in some frequency bands, customarily referred to as phononic gaps, which represent the mechanical equivalent of electronic and photonic bandgaps in semiconductors [8] and photonic crystals [9], respectively. While, in photonic crystals, band gaps originate from periodic variations of the refractive index, in PCs, they stem from the periodic modulation of the density and elastic modulus. Creating composite materials impeding or otherwise steering sound propagation in given frequency windows permits the manipulation of heat transfers in those windows [1]. In practice, to effectively impact heat transport, phononic devices must have spatial periodicity matching the nanometer wavelengths of terahertz phonons, their fabrication thus requiring the most advanced nanotechnology methods.

Recently, the scientific community has been shifting its main interest towards acoustic metamaterials (AMs) also due to a host of far-from-trivial and rather promising effects observed in these materials [10–13]. The main difference between PCs and AMs is that, while, in the former devices, sound propagation is manipulated via the structural periodicity, in the latter, local resonators [14,15] are used for the same scope. Structural periodicity might be advantageous, but it is not strictly required for the functioning of SMs, which makes the fabrication of AMs less challenging.

The works discussed in this review suggest that, as a potential alternative method to impact terahertz sound damping, one can include structural heterogeneities in a fluid, such as floating nano-objects. The mismatch of elastic properties between the floating colloids and the hosting liquid hinders the propagation of sound waves through the system, decreasing their lifetime. In this case, the arrangement of nano-objects is not present at all.

Regardless of the strategy adopted to impact sound propagation, a pivotal concern is the identification of best-suited characterization methods. Natural candidates are high-resolution inelastic X-ray (IXS) [16] or neutron scattering (INS) [17,18] techniques. Conceptually, an inelastic spectrometer resembles a microscope "pointed on the dynamics", which one can zoom in on dynamic events occurring in a given system over increasingly small distances and time lapses upon the increase of the energy, $E$, and the momentum, $\hbar Q$, exchanged in the scattering event; here, $\boldsymbol{Q}$ is the wavevector transfer and $\hbar = h/2\pi$ with $h$ the Plank constant. Notice that, due to the inherent isotropy of a disordered or a partially ordered system, the direction of the exchanged momentum is irrelevant, the parameter of interest being rather $Q = |\boldsymbol{Q}|$.

When dealing with a hybrid (liquid and solid state) system as an NP suspension, by increasing $Q$ and $E$, one could map the whole dynamic crossover between the hydrodynamic response and the intra-NP one. A host of recent IXS studies carried out by my research group focused on the dynamics of liquid suspensions along this crossover, demonstrating several non-trivial effects, including the ability of immersed NPs to decrease the lifetime and the amplitude density waves propagating through the hosting medium.

These findings have a broad general interest as, in principle, they may suggest a simple method to manipulate sound propagation in a simple liquid. More recently, a similar effect was observed in an IXS measurement on a frozen suspension, thus demonstrating that embedded nanoparticles can also impact the sound properties of solid-state aggregates. A critical review of these IXS investigations is the main focus of this paper. Before discussing these results, it is important to remark that, even at the lowest $Q$ values covered by these works, the probed phonon wavelengths were smaller than the NP size by almost an order of magnitude. Therefore, it is safe to assume that, in all cases, acoustic waves perceived the nanoparticles as infinitely massive, thus experiencing merely elastic collision at their interface.

## 2. The Pivotal Role of Bayesian Analysis

Overall, IXS studies on nanostructured materials are still sporadic, primarily due to the complexity of their dynamic response and the usually weak and poorly resolved inelastic contributions to their spectral profile. We recall here that the shape of an IXS spectrum from a Soft Matter system is often relatively unstructured. A typical example is displayed in Figure 1 and refers to an aqueous suspension of gold nanoparticles (Au-NPs) measured at $Q = 3$ nm$^{-1}$. The plot compares the raw measurement with its best fitting model line shape and its elastic and inelastic contributions. As typical of these measurements, the spectral features of the raw line shape are not well pronounced and mutually resolved. However, the comparison with the model profiles helps us recognize that, at these relatively low $Q$s, the spectral shape is somehow reminiscent of the well-known Brillouin triplet typically observed in the hydrodynamic limit. The latter features a central peak—usually dominant in an IXS spectrum from a disordered system—and two broad shoulders symmetrically sitting on its wings. The elastic peak arises from all non-propagating processes occurring in the system over timescales too long to be adequately resolved by the measurement, such as relaxation or diffusion. These processes affect density fluctuations in the target system, dominating them at low frequencies, where their essentially elastic contribution becomes preponderant over the inelastic one. This trend becomes more pronounced upon increasing $Q$, reflecting the decreasing ability of the system to support density wave propagation.

Aside from these merely qualitative aspects, any detailed modeling of the spectral shape would require a firm theory of the IXS line shapes in the mesoscopic regime. Unfortunately, such a theory is currently lacking and usually replaced by suitable phenomenological

generalizations of the exact hydrodynamic description valid at the lowest $Q$s and $E$s only. This generalized description demands, first and foremost, some assumption on the $Q$-dependent shape of the model profile and, in particular, on the number of spectral modes contributing to it. Aside from these inherent dilemmas, additional interpretation problems usually come from the $\chi^2$ best fitting routines involving too many adjustable parameters, whose result often turns out to be model-dependent.

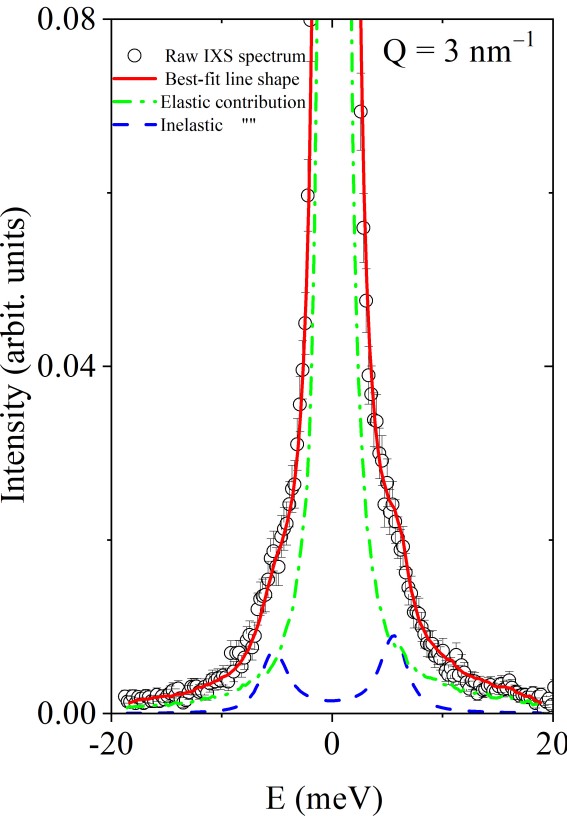

**Figure 1.** An example of an IXS spectrum from a dilute suspension of 15 nm sized gold nanoparticles, Au-NPs (circles), compared with the best fitting line shape and related elastic and inelastic contributions (see the legend).

To make the scenario even more puzzling, the insight typically sought from the line shape analyses of current IXS measurements is becoming increasingly informative and detailed. To unclutter this complicated matter, a reliable interpretation of the experimental spectral shape is highly desirable and would require, in the first place, evidence-based modeling recipes. In this endeavor, the use of Bayesian inference [19,20] is highly beneficial:

(1) It can rate the plausibility of competing model options;
(2) It privileges consistent models with fewer adjustable parameters;
(3) It protects the numerical algorithm from remaining trapped in local, rather than global, $\chi^2$ minima;
(4) It may be implemented using minimally invasive external constraints;
(5) It efficiently copes with the limited statistical accuracy of experimental results.

Among various topics, this review will illustrate how a Bayesian inference model enables an informative and unbiased interpretation of line shape measurements. However, before going any further, it is helpful to describe some general principles of the Bayesian inference approach followed in the data analysis presented here.

### 2.1. The Bayes Theorem

In the past, a few studies used Bayesian methods to extract accurate structure factor amplitudes from powder diffraction patterns with strongly overlapping Bragg peaks [20]. More recently, these methods have also been employed by my group to determine the number of excitations present in inelastic neutron or X-ray scattering spectra [21–24], or the number of diffusive processes in the frequency [25,26] or time-dependent [27] response of a system. The conceptual pillar of this approach is the Bayes theorem, whose enunciation also comes in handy here because it permits us to introduce the key distributions one deals with in Bayesian inference to model the measured spectral shape.

Suppose a spectral measurement is consistently described by a model identified by the set $\Theta = (\theta_1, \theta_2, \cdots, \theta_m)$, where the $\theta_m$ component of the vector $\Theta$ is the $m$th parameter defining the spectral profile, as, e.g., the amplitude, inelastic shift, or damping of a mode, the background, and similar.

If we then represent the measurement outcome with the vector $(y_1, y_2, \cdots, y_n)$, where $n$ is the number of sampled values, we can assume that the latter provide a sampling $y_i$ ($i = 1, \cdots n$) of a random variable $y$. We can also formally represent our a priori information on the physical problem under scrutiny with the symbol $I$. With this notation, the Bayes theorem [19] reads as:

$$P(\Theta|y, I) = \frac{P(y|\Theta, I)P(\Theta|I)}{P(y|I)},\tag{1}$$

where $P(\Theta|y, I)$ is the posterior distribution of the model parameters conditional on the achieved measurement and the available information. Our a priori knowledge—or lack thereof—about the model parameters is reflected by the prior distribution $P(\Theta, |I))$. The distribution $P(y|\Theta, I)$ is the likelihood of the data and measures the plausibility of a given outcome conditional on the truth of the information and the model. Finally, $P(y|I)$ is the marginal probability, whose primary role is enabling the probability in Equation (1) to be correctly normalized, i.e., to have the unit integral over the integration variable $\Theta$. Let us now comment in further detail on these distributions, starting with the prior distribution (or simply the prior).

### 2.2. The Prior Distribution

As mentioned, the prior probability incorporates all prior knowledge on the model, including physical constraints, sum rules, or previous results it needs to comply with. This previous knowledge can be accrued, e.g., from the general line shape theory, a computation, a measurement, or any relevant direct or indirect literature finding. The inclusion of the previous information in the inference process is peculiar to the Bayesian approach and can be more or less coercive. For instance, a specific parameter, $\theta$, must attain a precise value because it was already measured or computed in previous works. We may thus assume that the parameter takes that with unit probability. Suppose we opt for a more agnostic approach. We can, therefore, impose that the parameter follows a Gaussian distribution centered at $\theta^\star$ with variance suitably chosen to limit parameter swings around the expected value. By doing so, we assign a higher probability to parameter values in the neighborhood of $\theta^\star$.

The information available on the parameters might be more vague. For instance, we might only know that a specific parameter must be non-negative or vary in a limited interval, thus requiring a truncated distribution. Parameters having an a priori unknown value inside a domain can be assumed to be uniformly distributed within such a domain. Also, whenever feasible, any mutual entanglement between parameters and any selection, conservation, or sum rule should be translated in a workable analytic form into the prior distribution function.

### 2.3. The Likelihood Function

The likelihood is the joint probability of obtaining a specific measurement outcome, conditional on the accuracy of the adopted model. In other words, it evaluates how plausible a particular result is, assuming the model adopted is correct. As mentioned, the set of measurements $y = (y_1, y_2, \cdots y_n)$ can be interpreted as the sample of a random variable $Y_i$, which follows the (parent) distribution $f(Y_i; \Theta)$. The main grounds of this interpretation are that, after repeating a data collection, one will generally obtain a different result, even under the same experimental conditions. After data collection, $y_i$ becomes a specific realization (sampling) of the random variable $Y_i$. Since the variables $Y_i$ are independent, the compound probability of the whole sampling is the product of the probabilities of the individual values $y_i$. Our spectroscopy results read as follows:

$$y_i = S_M(Q, E_i) + \epsilon_i, \tag{2}$$

where $S_M(Q, E)$ is the model that depends on a vector of unknown parameters $\Theta$, while $\epsilon = (\epsilon_1, \epsilon_2, \cdots, \epsilon_n)$ is a vector of random errors. The latter can be assumed independent and normally distributed; hence, the sampling $\epsilon_1, \cdots, \epsilon_n$ has probability distribution:

$$P(y|\Theta) = \prod_{i=1}^{n} \frac{1}{\sqrt{2\pi\sigma_i^2}} exp\left[-\frac{[y_i - S_M(Q, E_i)]^2}{2\sigma_i^2}\right] = const \cdot exp\left[-\sum_{i=1}^{n} \frac{[y_i - S(Q, E_i)]^2}{2\sigma_i^2}\right]. \tag{3}$$

Notice that the quantity above can be interpreted as the probability of obtaining the sampling $y_i$ conditional on the model's accuracy, which, by definition, is the likelihood of the whole set of $N$ values $y_i$.

### 2.4. The Posterior Distribution and Its Normalization

The quantity on the left-hand side of Equation (1) is the joint posterior distribution of the model parameters. This distribution takes into account both the prior knowledge and the measurement outcome. Any Bayesian inference rests on this probability distribution, which incorporates the earlier knowledge—through the prior—and the data evidence—through the likelihood function. In summary, one can recognize that the final posterior distribution is obtained by updating the previous knowledge with the data evidence, i.e., the measurement outcome.

To obtain estimates for a single parameter $\theta_k$, one needs to integrate the posterior distribution over all other parameters, except $\theta_k$. This integration is often referred to as marginalization, the marginalized distribution being denoted as $P(\theta_k|y) = \int_{\Theta_{-k}} P(\Theta|y)d\Theta_{-k}$. Here, the integrand $\Theta_{-k}$ includes all components of the parameter vector except $\theta_k$. To estimate the most plausible value of $\theta_k$ and its uncertainty, one can consider the mean and the standard deviation of $P(\theta_k|y)$, respectively. Consistently, the probability that the parameter $\theta_k$ belongs to a specific interval is evaluated by integrating its marginal posterior over such an interval.

Similarly, we can introduce a marginal likelihood, representing the probability of achieving the measurement $y_i (i = 1, \cdots n)$, irrespective of the model parameters, i.e., integrated over all of them. After this integration, the likelihood becomes the constant—$P(\Theta|I)$—required to normalize the posterior parameter distribution $P(\Theta|y, I)$ in Equation (1). Unfortunately, no analytic expression of $P(\Theta|I)$ is generally available to make such an integration explicit, and even if it were, the multidimensional integral to be performed on it would have likewise been overly complicated to compute. Therefore, the posterior distribution is usually drawn up to a normalization constant. In this endeavor, the use of the Monte Carlo Markov chain (MCMC) comes in handy, as it can simulate the joint posterior distribution, even if the normalization constant is unknown. Inference is then made on this simulation rather than a close analytical form of the posterior distribution. The reader can find further details on this method in Refs. [28,29].

### 2.5. Occam's Principle

Even though Bayesian and classical analyses give consistent results in the asymptotic limit of an infinitely large sampling, the two approaches are not equivalent. Indeed, the former always provides the advantage of drawing probability distributions rather than just optimal parameter values. This feature is especially advantageous when comparing the performance of competitive models. Perhaps more importantly, the Bayesian method embodies the Occam's razor principle, which states that, among competing hypotheses (models) satisfactorily explaining some evidence, the one with the smallest number of adjustable parameters is preferable. This parsimony principle is inherent to the Bayesian theorem in Equation (1). In fact, in its explicit form, the parameter posterior contains the product of prior parameter distributions, which, by definition, have values less than or equal to one. Therefore, adding parameters to the model amounts to multiplying the posterior (of each parameter) by additional factors less than or equal to one, thus correspondingly lessening the parameter's posterior, i.e., making any guess on such a parameter less plausible.

### 2.6. The MCMC Method

In practice, MCMC methods construct an ergodic Markov chain that draws $\Theta^m$, with $m = 1 \cdots M$, and a stationary distribution corresponding to the joint posterior distribution. The number of updates of the parameter values in an MCMC algorithm is generally called the number of sweeps, denoted by $M$. A new draw of the posterior is obtained at each algorithm sweep. The latter updates all the parameters in sequence, drawing each from the respective posterior distribution, conditional on the other parameters' values. Further details on the algorithm moves can be found in Refs. [21,30].

I finally stress that the Bayesian approach described here is ideally suited for minimally biased and evidence-based modeling of experimental results. Upon suitable modifications, it can be applied to an unlimited class of measurements in which the resolved variable can be the energy, as in inelastic scattering; the time, as in time-resolved measurements; the angle, as in X-ray and neutron diffraction; or any other relevant probe parameter scanned. Furthermore, Bayesian methods are equally valuable when modeling the response of a measurable quantity to the variation of temperature, pressure, or any other sample conditions. Unlike frequentist approaches, whose rigorous predictions heavily rest on the asymptotic sampling, i.e., the unlimited repetition of a measurement, Bayesian inference enables robust inference from single-event experiments, which are very common in modern science.

## 3. What Can We Learn from an IXS Measurement on a Suspension?

An extended treatment of the theoretical and practical aspects of the IXS technique goes beyond the scope of the present review, and the interested reader is deferred to the existing literature [16,31]. Here, I recall some very general features we could expect from an IXS measurement on a simple system.

As discussed in the introductory section, an inelastic scattering spectrometer can focus on dynamic events occurring over different time and length scales upon suitable $Q$ and $E$ variation. In principle, this strategy could enable the whole mapping of dynamic crossover between the hydrodynamic and the single-particle regimes. In practice, however, this mapping cannot be achieved using a single X-ray spectrometer, as the access to hydrodynamic scales would require using visible or UV light as a probe. Nonetheless, at the lowest $Q$s reachable by IXS, the dynamic response of a liquid is averaged over sufficiently long distances and time for a suitably generalized hydrodynamic description to hold validity. Evidence of the transition of the IXS spectrum from this generalized hydrodynamic regime to the single-particle one was provided, e.g., in an IXS work on lithium [32], whose main result is summarized in Figure 2. At the lowest $Q$s, the spectral shape consists of a relatively sharp triplet profile dominated by the generalized hydrodynamic modes. At low $Q$, the inelastic shift of the two side peaks increases (linearly) with $Q$, as suggested by the dashed line roughly connecting their maxima in the Stokes sides. As $Q$ increases, the overall

spectral shape evolves toward the Gaussian profile featuring the single-particle response. In this regime, the probed event essentially reduces to the free recoil of the single struck atom induced by the collision with the impinging probe particle, such as a photon, in the case of IXS.

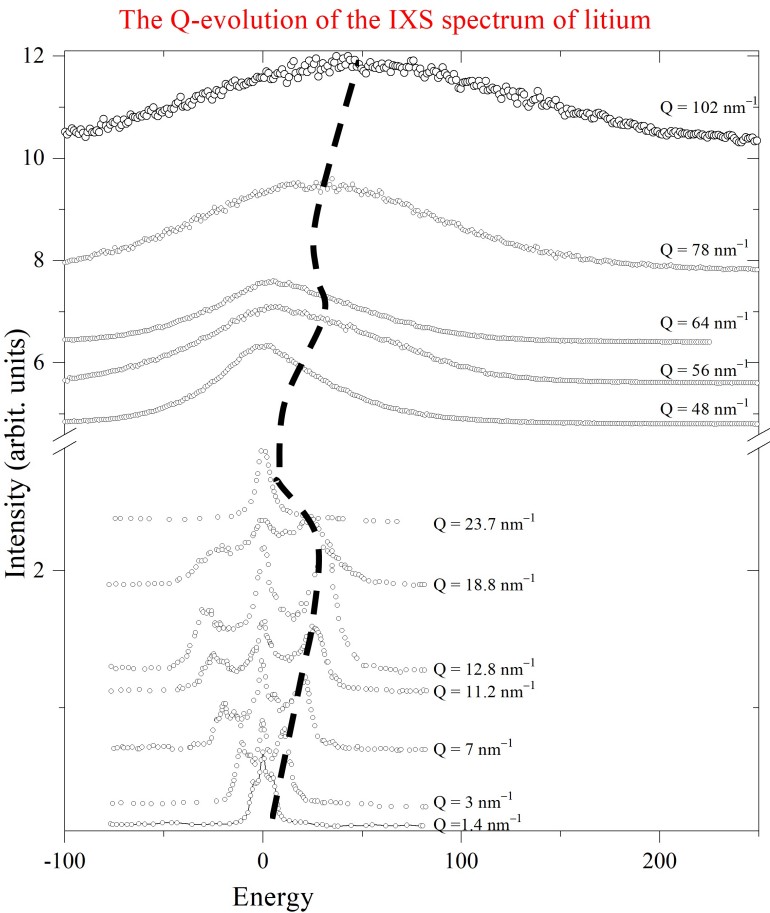

**Figure 2.** The *Q*-transition of the IXS spectral shape of lithium (open circles) from the quasi-hydrodynamic regime (bottom) to the single-particle one (top). The spectra are collected at the indicated *Q* values and vertically offset for clarity. The dashed line is a guide to the eye that indicates the position of collective inelastic excitations in the spectrum. The spectral line shapes are from Ref. [33], while the whole figure is taken from Ref. [16].

Similarly, probing the spectrum from a suspension at different *Q*s would enable a mapping of the dynamical processes occurring in the sample over different scales, as schematically illustrated in Figure 3.

Suppose the measurement is performed at low *Q* values well below the IXS range. In this case, if the immersed NPs interact, one would expect inter-NP phonons to become a relevant part of the system's response over these large distances. These phonon modes consist of density fluctuations originating from displacements of the NPs from their equilibrium positions, which transmit from site to site due to the NPs' interactions. In principle, these modes overlap with collective excitations propagating through the hosting liquid and, more generally, with any other dynamic processes in such a liquid, such as relaxations and diffusions.

At larger *Q* values, one should expect the emergence in the spectrum of phonons propagation modes inside the NPs (intra-NP phonons).

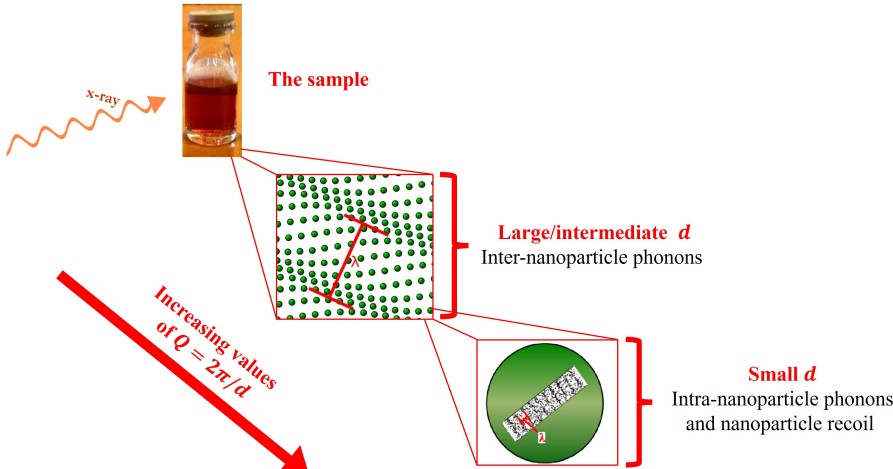

**Figure 3.** A schematic representation of what one would expect to observe by performing an IXS measurement on an NP suspension for increasing $Q$ values. The figure is taken from Ref. [22].

The case of a dilute suspension is more straightforward than the one discussed above, as the long NP separations eliminate inter-NP phonons. Furthermore, the sparse character of the suspension likewise translates into a dynamic response similar to that of the pure hosting liquid. More interestingly, NP suspensions are simple prototypical systems ideally suited to investigate the manipulation of the acoustic response of a fluid via the inclusion of structural heterogeneity. It is also worth noticing that dealing with small concentrations in metal NP suspensions is imperative, as the requirement of sample stability limits the concentration; indeed, the average inter-particle distance must be long enough to prevent the quick formation of aggregates. Although our group investigated the spectral response of a higher concentration NP suspension, the study focused on silica NPs, for which electrostatic interactions were not a concern. However, any assessment of the effect of NP concentration was inconclusive. Still, no IXS measurements have been reported on these systems until less than a decade ago. In recent years, my research group has intensively investigated the dynamics of diluted NP suspensions to shed some light on this uncharted field.

## 4. Experimental Details

### 4.1. The Measurements

We utilized the high-resolution IXS beamline Sector 30 of the Advanced Photon Source at Argonne National Laboratory [34,35] to measure all suspension spectra presented in this paper. The sample scattering was energy-analyzed by nine independent analyzers. These analyzers were placed on the moving extreme of a spectrometer arm. The latter can be rotated in the horizontal plane to select the exchanged wavevector through $Q = 2\pi/\lambda sin\theta$, where $\lambda$ and $2\theta$ are the incident beam wavelength and the scattering angle, respectively. The spherical analyzers were placed at a constant angular offset corresponding to a $Q$ separation of 2 nm$^{-1}$. The incident beam energy was 23.7 keV and corresponded to the Si(12 12 12) backscattering reflection from the analyzers, while the energy analysis was implemented through the rocking of the crystals of the monochromator unit while keeping the analyzers fixed. The shape of the energy resolution profile slightly varied for each analyzer, with an average spectral width of about 1.2 meV. More details on the spectrometer are given in Ref. [34].

### 4.2. The Line Shape Model

The model used to describe the structure factor is the sum of a finite number of excitations yielding the following analytical profile:

$$S(Q, E) = A_e(Q)\delta(E) + [n(E) + 1]\frac{E}{k_B T}\left\{ L_{A_0, z_0}(Q, E) + \right.$$

$$\left. \sum_{j=1}^{k} \frac{2}{\pi} A_j(Q) DHO_j(Q, E) \right\}. \tag{4}$$

Here, the term $\delta(E)$ is the Dirac delta function, which defines the elastic component of the spectrum having amplitude $A_e(Q)$. The energy-dependent term $n(E) = (e^{E/k_B T} - 1)^{-1}$ is the Bose factor, which ensures the fulfillment of the detailed balance condition. The term in curly brackets is the sum of a Lorentzian central peak having half-width at half-maximum $z_0$ and amplitude $A_0$. This term accounts for the spectral contribution of non-propagating dynamical processes in the sample, such as diffusion and relaxation phenomena. The $k$ inelastic contributions are described by Damped Harmonic Oscillator ($DHO_j(Q, E)$) terms having amplitude $A_j(Q)$. The parameters defining the shape of the DHO excitations, $\Omega_j(Q)$ and $\Gamma_j(Q)$, represent the undamped energies and damping coefficients, respectively, they determine the analytical profile of the $i$th excitation through the following:

$$DHO_j(Q, E) = \frac{\Omega_j^2(Q) * \Gamma_j(Q)}{(E^2 - \Omega_j^2(Q))^2 + 4[E\Gamma_j(Q)]^2} \tag{5}$$

It is worth noticing that the number, $k$, of $DHO_j(Q, E)$ excitations and their shape coefficients are equally treated as adjustable model parameters.

Finally, to provide an accurate approximation of the measured line shape, the model function in Equation (4) should be convoluted with the instrument resolution function $R(E)$ and the result summed to the spectral background. Explicitly:

$$\tilde{S}(Q, E) = R(E) \otimes S(Q, E) + B(E) \tag{6}$$

where $B(E)$ is a mildly $E$-dependent background intensity.

### 5. Results

*Intra-NP Phonon Excitations*

The first observation of phonon modes propagating through the NP interior dates back to our IXS study on an aqueous suspension of gold nanoparticles (Au-NPs) [22]. An important finding of such a work is illustrated by Figure 4, which compares two IXS spectral shapes measured at low $Q$s with the respective best fitting model profiles, along with their two inelastic contributions, determined through the Bayesian inference method described in this paper. We assigned the two inelastic features to acoustic modes of either transverse or longitudinal polarization propagating inside the Au-NPs' interior. The spectrum of pure deuterated water measured in a previous joint IXS and neutron scattering work [36] is also reported in the plot for reference. It is evident that the two broad inelastic shoulders present in the water spectrum essentially disappear upon NP immersion and are replaced by the two small peaks representative of intra-NP phonons. This finding might seem surprising, considering that the dilute nature of the suspension should make the intensity contribution from the gold dynamics relatively small. Indeed, their visibility highlights the mentioned ability of immersed NPs to damp the acoustic modes of the hosting medium.

In Figure 5, the $Q$ dependence of the phonon's inelastic shifts, i.e., the sound dispersion curve, is reported as derived through the Bayesian analysis. Our assignment of the two peaks to Au phonons is further supported by the favorable comparison between the dispersion curves and the acoustic branches of gold, as illustrated in Figure 5. Most importantly, the maxima of the two dispersion branches roughly match the energy of the peaks of the density of state of bulk gold, which is reported on the right as computed in Ref. [37].

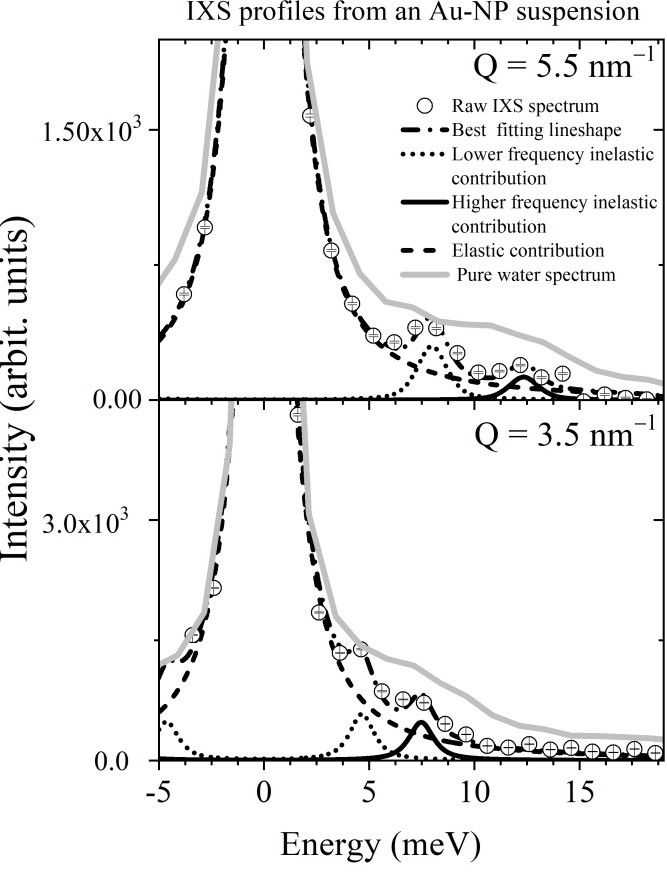

**Figure 4.** Two IXS spectra collected at the indicated *Q* from a dilute aqueous suspension of gold nanoparticles (Au-NPs) are compared with the best fitting model line shapes determined through the Bayesian inference method discussed in the text. The IXS spectra of pure water collected at similar *Q* values are also reported for reference and vertically offset for clarity. All data are from Ref. [22].

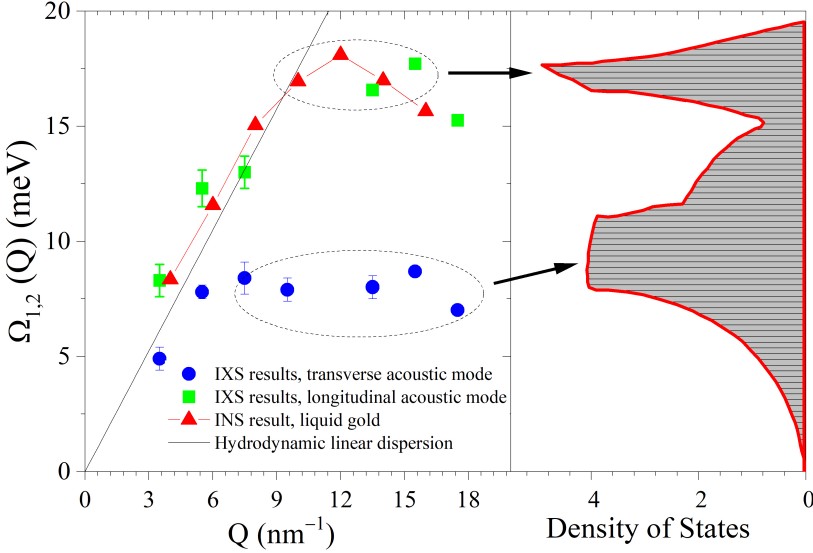

**Figure 5.** (**Left panel**) The dispersion curves of the inelastic modes derived from the IXS spectra of the Au-NP suspension (see Figure 4) through the Bayesian inference analysis, as indicated in the legend. Results on liquid gold from Ref. [38]. (**Right panel**) The density of state (DoS) of bulk gold is reported as computed by the lattice dynamics in Ref. [37]. The arrows emphasize the rough correspondence between the dispersion curve maxima and the main DoS features. Data are redrawn from Ref. [22].

## 6. Using Nanoparticles to Damp High-Frequency Sound Waves in a Fluid

As suggested by the above discussion, the most noticeable difference between the line shape of a diluted NP suspension and that of the pure solvent is that the inelastic modes of the hosting liquid are highly attenuated in the former profile. We achieved the most striking evidence of this effect when investigating the dynamic response of an Au-NP suspension in glycerol [39], whose IXS spectra are compared to the ones of pure glycerol in Figure 6. The pronounced inelastic shoulders in the pure solvent spectra are hardly visible in the suspension ones, which have inelastic wings barely emerging from the spectral background.

At this stage, there is still a possibility that the comparison shown in Figure 6 is misled by the normalization to the elastic ($E = 0$) maximum. One can argue that, since this maximum is more intense in the suspension spectrum, normalizing to its height may cause an apparent, but not particularly significant, attenuation of the inelastic shoulders.

Figure 7 provides an answer to this legitimate doubt. Data in the plot were derived from Ref. [40] and suggest that, aside from the relative amplitude reduction, the dominant inelastic mode presents a significant increase of the relative damping, as defined by the ratio $\Gamma/\Omega$ between acoustic damping and frequency.

To understand this effect, it is helpful to recognize that, in the mesoscopic regime probed by IXS, the inequality $Qd_c/2\pi \gg 1$ (with $d_c$ being the colloid diameter) remains valid even for nm-sized colloids. At these scales, multiple scattering significantly impacts acoustic propagation, as acoustic waves are more prone to get reflected at the surface of the colloid [41]. These reflections prevent sound waves in the surrounding liquid from entering the colloid and vice versa. Due to multiple interface reflections, there is a considerable mutual dephasing between acoustic waves, leading to a significant enhancement in acoustic damping caused by interference.

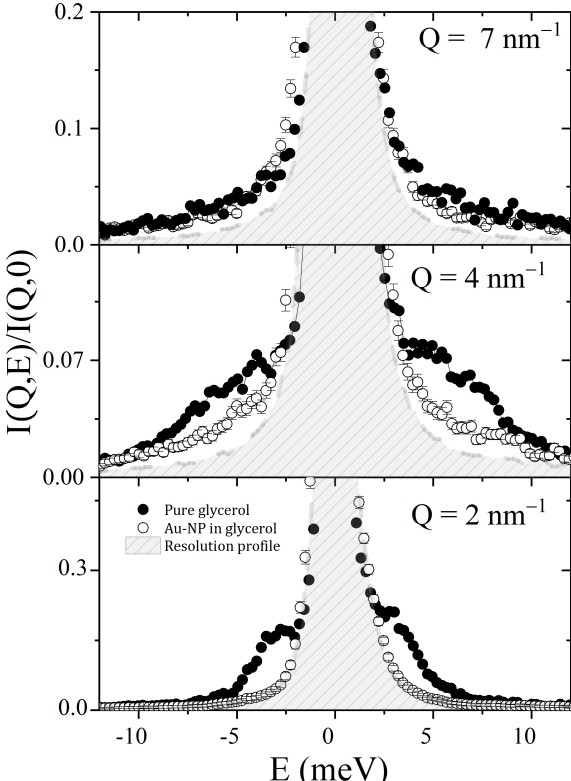

**Figure 6.** The IXS scattering profiles from a suspension of Au-NPs in glycerol and pure glycerol collected at the indicated $Q$ values are reported after normalization to the respective maxima. The corresponding resolution profiles are also included for reference after similar normalization. The figure was reproduced from Ref. [22].

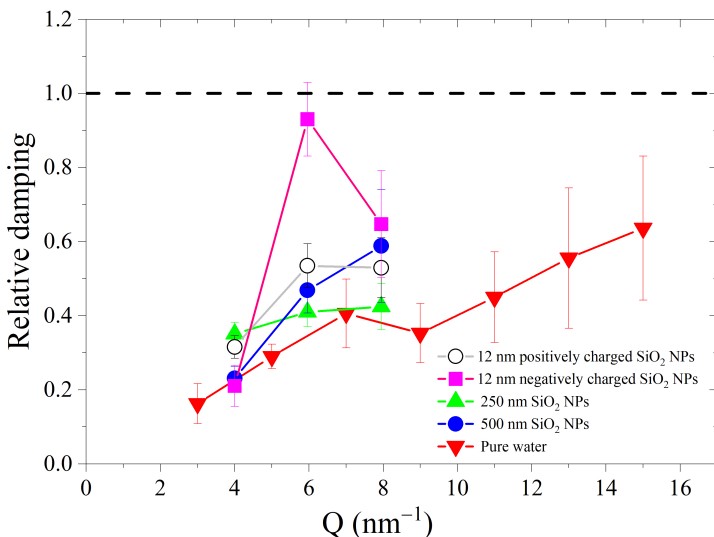

**Figure 7.** The relative damping of the dominant inelastic mode in several suspensions of either charged or neutral SiO$_2$ NPs of various sizes (see legend) is reported as a function of $Q$. The dashed line shows the critical damping condition. The data are from Ref. [40].

Another intriguing effect connected with the enhanced interface reflection is the appearance of an additional mode resulting from the propagation of interface density waves, customarily known as Stoneley waves [42]. An IXS work on glycerol documented the fingerprints of these waves in the terahertz spectrum of glycerol. Their primary signature is a low-energy branch with a nearly linear $Q$ dependence, as shown in Figure 8.

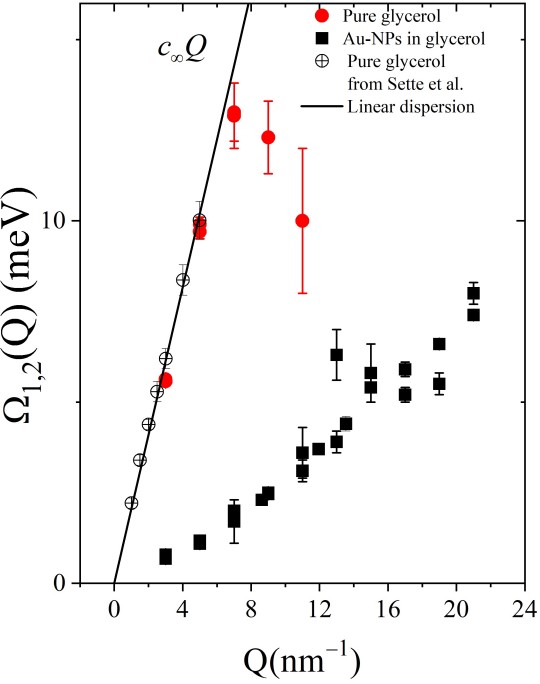

**Figure 8.** The plot displays the dispersion curves determined through the Bayesian analysis of the IXS spectra of an Au-NP suspension in glycerol. Red dots and black squares refer to the longitudinal acoustic mode and the interfacial mode, respectively (see the text). The data are from Ref. [43] and are compared with the IXS measurement on pure glycerol by Sette and collaborators (Ref. [44]). The linear dispersion representing the high-frequency elastic response of glycerol is also reported as a solid line for comparison.

At this stage, a still unanswered question is whether a damping effect similar to the one observed in dilute liquid suspensions can also be observed in a more rigid and ordered system, such as a crystal with NPs embedded. A clear advantage in dealing with a non-amorphous solid is the sharpness of its phonon features, which makes any possible line shape modification induced by NPs more easily discernible. In a recent IXS work [45], we addressed this issue by investigating the phonon spectrum of a frozen aqueous suspension of Au-NPs. We found that, even at a low concentration (about 0.1% in volume), embedded NPs substantially impact the phonon spectrum of the hosting medium, at least at some $Q$ values. In fact, after looking at Figures 9 and 10, one readily notices that the embedded nanoparticles significantly attenuate the dominant phonon peak of ice (at about 8 meV). This conclusion stems from comparing the non-normalized line shapes, suggesting that the inclusion of NPs causes an attenuation of the phonon mode at about 8 meV, which is more pronounced than that experienced by the elastic peak of the same spectrum. This trend is unlikely to be a unique consequence of the more significant absorption of embedded NPs, as any absorption increase would have affected all spectral modes, elastic or inelastic, in equal measure.

Furthermore, the line shapes in the left plot of Figure 9 exhibit a side feature in the 9.5–12 meV range (encircled by the ellipsis), whose intensity is nearly the same in the ice and suspension spectra despite the different absorption of the two samples. However, in ice, it is shifted at a slightly larger energy transfer. In summary, the comparison in Figures 9 and 10 evidences distinctive features not trivially amenable to the X-ray absorption increase caused by embedded NPs or their additional, intra-NP phonon modes. Therefore, we can conclude that the impact of embedded NPs on the phonon spectrum of a solid is a genuine effect, which, however, seems to affect the various spectral modes to a different extent in a way not very well understood at present.

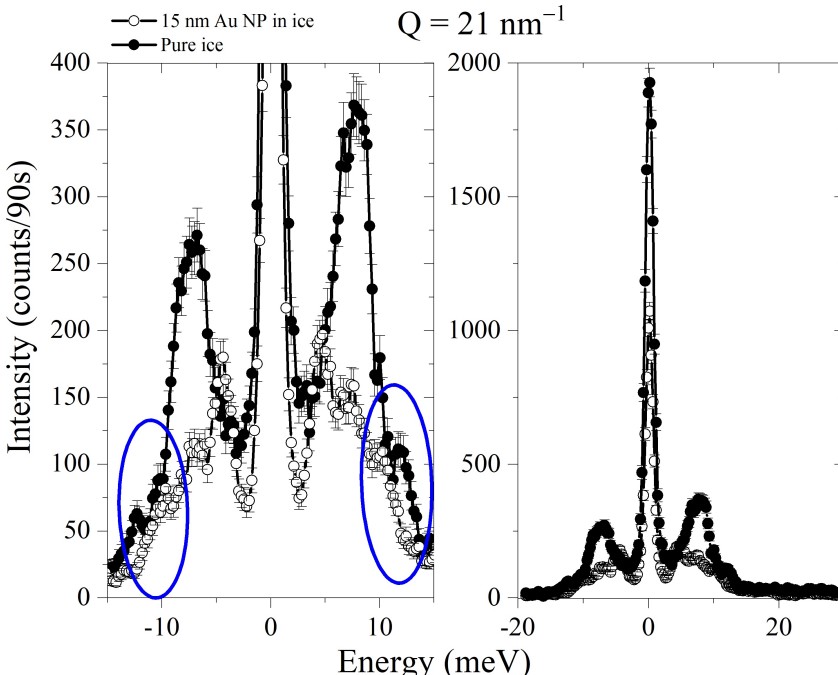

**Figure 9.** (**Left panel**) Comparison between the IXS spectrum measured at Q = 19 nm$^{-1}$ of the ice sample and the frozen suspension. The spectral profiles are displayed in an expanded view, and the dashed ellipses encircle the regions where the difference between the phonon modes of the two samples is most evident. (**Right panel**) The same comparison is reported at full scale.

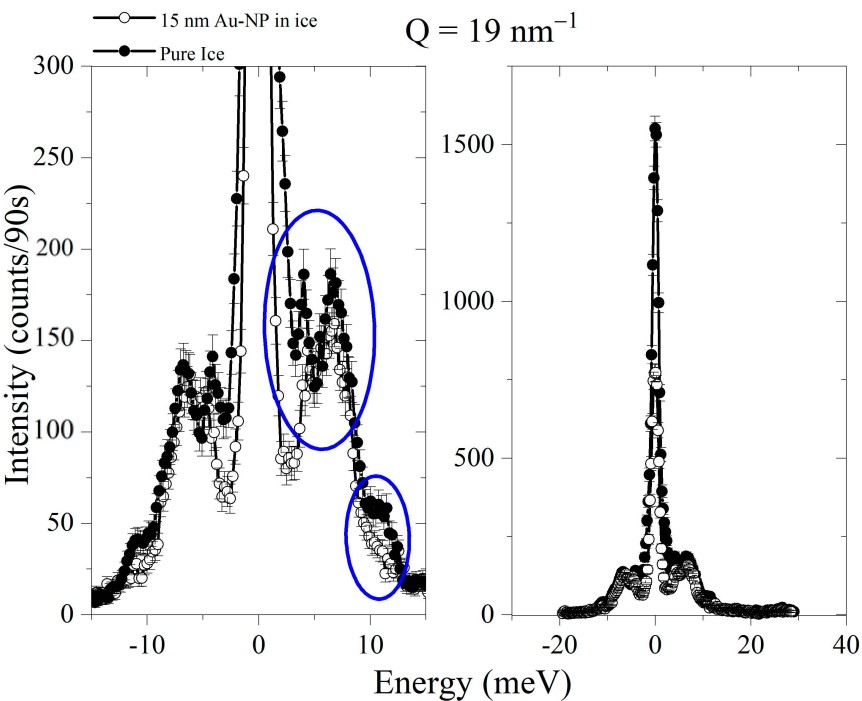

**Figure 10.** The same as Figure 9, but for $Q = 21$ nm$^{-1}$.

*The Bayesian Inference at Work: Two Examples*

Let us briefly discuss how Bayesian inference can help with line shape analysis and interpretation.

To provide a practical example, the typical outcome of the Bayesian analysis of the *IXS* spectra from a nanoparticle suspension is summarized in Figure 11, which refers to the IXS results discussed in Ref. [22]. In Panel A, I compare the *IXS* spectra from an aqueous suspension of gold NPs, the corresponding best fitting model line shapes, and their spectral components, all already displayed in Figure 4.

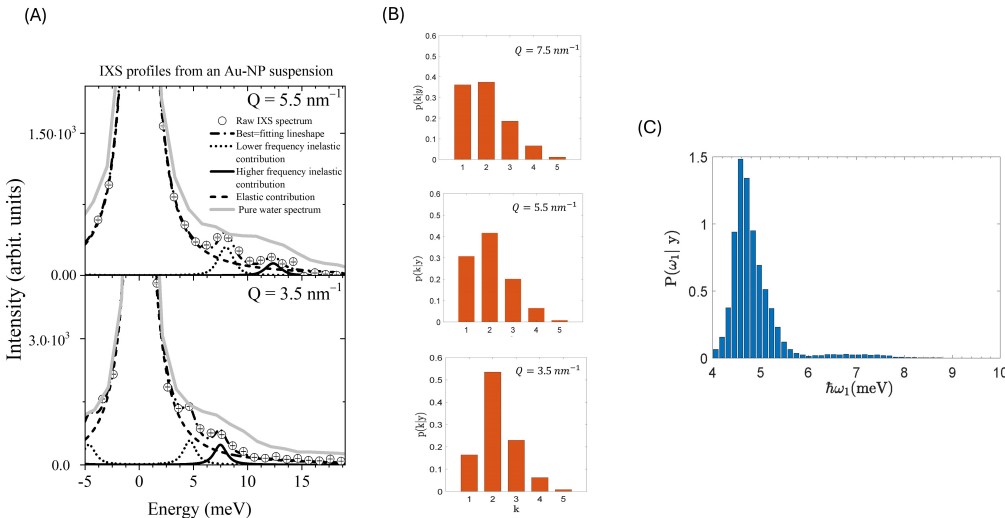

**Figure 11.** (**Panel A**) displays the same spectral line shapes already shown in Figure 4. (**Panel B**) shows the posterior distribution of the number of inelastic components drawn in the same IXS study for three $Q$ values. (**Panel C**), finally, illustrates the posterior of the inelastic shift of the lower frequency phonon mode for the $Q = 3.5$ nm$^{-1}$ spectrum.

Panel B shows that the relative probability of the most plausible model option ($k = 2$) decreases upon an increase in $Q$, while the probability of the alternative model option ($k = 1$) correspondingly increases. Most importantly, the sharp and unimodal posterior distributions of the inelastic shifts (e.g., that in panel C) support the physical significance of the two spectral modes detected and the robustness of the related line shape model.

A second noticeable example of the outcome of a Bayesian analysis can be found in Figure 12, which refers to our INS work on liquid silver. There, we used Bayesian inference to settle a controversy about the presence of a second spectral excitation of a transverse character. The plot displays the posteriors $P(\Omega_l|y)_k$ of the $l$-th mode's inelastic shift drawn for a $k$ excitation model option and for the IXS spectrum at $Q = 16$ nm$^{-1}$. The Bayesian algorithm identifies the two most plausible models as those corresponding to either $k = 1$ or $k = 2$. It appears that, within the validity of the $k = 2$ model, the second low-frequency excitation has an extremely broad and flat posterior, as opposed to the sharply peaked one of the higher frequency excitations. Therefore, the evidence for the second (controversial) excitation appears weak. Of course, the investigator is the only one ultimately entrusted with the final decision on the number of modes present in the spectrum. However, Figure 13 speaks against the robustness of a double-mode hypothesis. As a plausible alternative, the low-frequency excess of the spectral intensity could have been ascribed instead to a viscoelastic dynamic behavior due to an active structural relaxation.

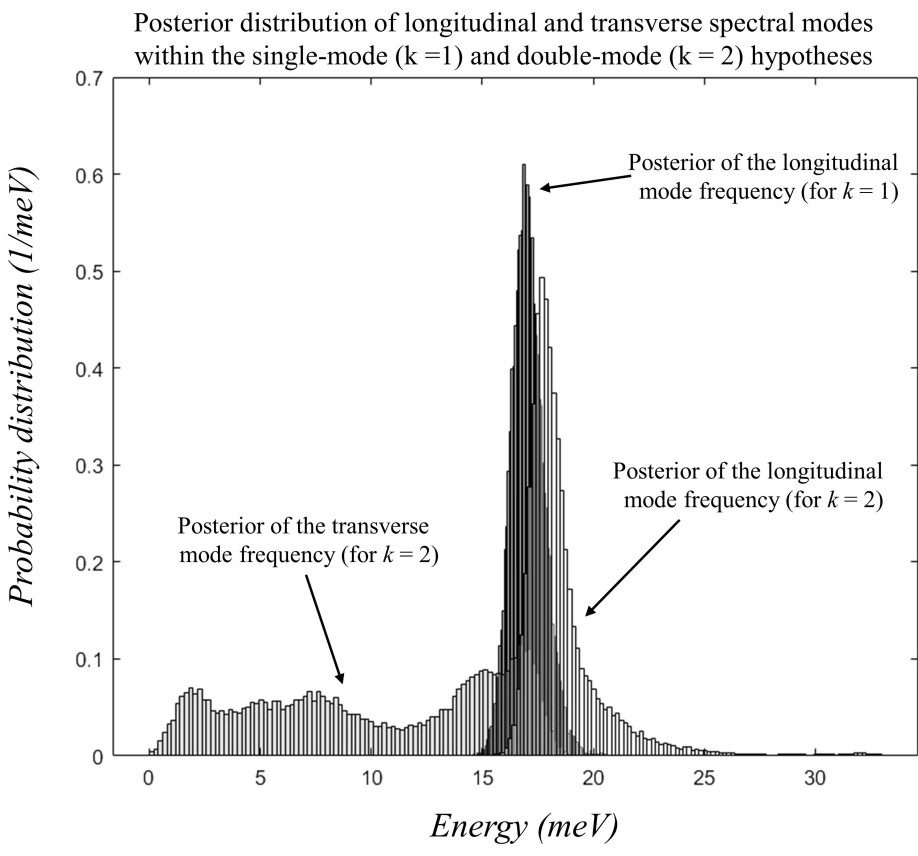

**Figure 12.** Three posterior distributions of the inelastic shift $\Omega_l$ are reported as derived from the Bayesian inference analysis of the INS spectrum of liquid silver at $Q = 16$ nm$^{-1}$. They refer to the alternative model options, including either one ($k = 1$) or two ($k = 2$) inelastic modes in the spectral line shape. The data are from Ref. [46].

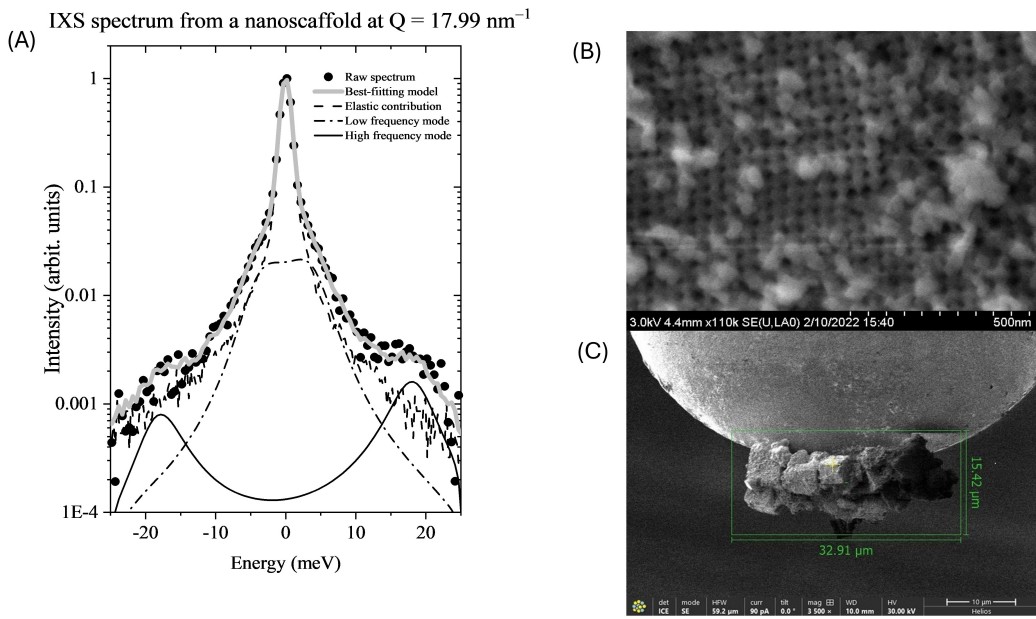

**Figure 13. (Panel A)** IXS spectrum of a $SiO_2$ nanoscaffold compared with the best fitting line shapes (see the legend). **(Panel B)** High-resolution electron microscopy image of one of the silica nano-platforms used as a sample for the IXS measurements (see text). **(Panel C)** Microscope image of the nano-scaffold assembly and related holder used to perform the measurement.

## 7. Looking Ahead

As an extension of the studies discussed in this paper, one could consider systems of NPs interacting in a controlled fashion. In this respect, at Brookhaven National Lab, a group led by O. Gang developed a new method to assemble nanoparticles (NPs) into ordered structures using polyhedral DNA frames [47]. These structures can be integrated by encapsulating NPs with grafted DNA strands [48]. The symmetry of the vertices determines the structure's coordination and overall geometry, while the linkage scaffold is uncoupled from the characteristics of the NPs embedded in it. The 3D DNA frameworks can be "converted" by promoting inorganic material growth on DNA struts. This procedure allows for the creation of several inorganic lattice frameworks made of silica [49], superconducting niobium [50], silicon carbide [51], and more. The size of the voids can be regulated through the growth time, allowing control of the scaffold's void ratio from 20% to 100% (bulk solid). Preliminary measurements were performed on these systems at APS-Argonne National Laboratory and NSLS-II-Brookhaven National Laboratory using inelastic X-ray scattering. The main result of such a measurement is summarized in Figure 13. Although the features observed in the measured line shape look promising, the scattering signal still needs sensible improvement. Specifically, an enhancement in the statistical accuracy and signal-to-background ratio would require larger samples and optimized sample+substrate arrangements.

**Funding:** This research received no external funding.

**Institutional Review Board Statement:** Not applicable.

**Informed Consent Statement:** Not applicable.

**Data Availability Statement:** The original contributions presented in the study are included in the article, further inquiries can be directed to the corresponding author.

**Acknowledgments:** I warmly acknowledge A. De Francesco for the help with the figure production and the stimulating discussions.

**Conflicts of Interest:** The author declares no conflicts of interest.

## Abbreviations

The following abbreviations are used in this manuscript:

| | |
|---|---|
| AM | acoustic metamaterial |
| APS | advanced photon source |
| Au-NP | gold nanoparticle |
| E | energy transfer |
| INS | inelastic neutron scattering |
| IXS | inelastic X-ray scattering |
| MCMC | Markov chain Monte Carlo |
| NP | nanoparticle |
| PC | phononic crystals |

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
