# Peer review of "Inelastic X-ray Scattering as a Probe of Terahertz Phonon Propagation in Nanoparticle Suspensions"

_applsci, doi:10.3390/app14083377_

Round 1

Reviewer 1 Report

Comments and Suggestions for Authors

Reviewer 2 Report

Comments and Suggestions for Authors

The paper reviewed recent inelastic X-ray scattering on nanopatical suspesions and highligted Bayesian analysis on the results. The overall review is good but need to be improved for publishing.

1. There are a lot of typos around Figure4. Some figures are labeled as Fig.??. Also, there is a loss of reference in the figure caption.

2. The authors should give an example of suspensions with different concentration and discuss more about the concentration related scattering results.

3. The authors should also mention the application of Bayesian analysis on other curve fitting, not only limited to inelastic X-ray scattering.

Reviewer 3 Report

Comments and Suggestions for Authors

Dear author, thank you for sending this manuscript which is interesting to be published in Applied Sciences. There are minor changes that should be adjusted before publication, detailed as follows:

- the abstract contents should be improved, by anticipating the results of your study

- figure 2 is of low quality: please improve the saving format (tiff is usually one of the best)

- please add the legend to all graphs; description of lines in the caption is not sufficient. the image should be self-explanatory;

- figure 3 is of low quality: a screen shot is not sufficient for scientific publications

- in figure 4 there are markers missing from the legend, referring to the non-filled circles. imagine to print b/w, are all the squared markers helping the reading? please use circles, triangles, stars, ... as markers.

- figure 6: change markers or line style

- figure 12: low quality

- references is written twice

There is a major change that should be carried out as priority:

- there are several citations to the same authors (i.e. De francesco et al.). this is not good and the citations should be varied, also because some of them are referring to the same topic.
